# Analysis of the Behavior of SMA Mixtures with Different Fillers Through the Semicircular Bend (SCB) Fracture Test

**DOI:** 10.3390/ma12020288

**Published:** 2019-01-17

**Authors:** Pedro Limón-Covarrubias, David Avalos Cueva, Gonzalo Valdés Vidal, Oscar Javier Reyes Ortiz, Rey Omar Adame Hernández, José Roberto Galaviz González

**Affiliations:** 1Department of Civil Engineering and Topography, Guadalajara University, Guadalajara 44430, Mexico; ingenieria_limon@hotmail.com (P.L.-C.); galaviz.gonzalez.r@gmail.com (J.R.G.G.); 2Department of Civil Engineering, Universidad de La Frontera, Temuco 01145, Chile; gonzalo.valdes@ufrontera.cl; 3Department of Civil Engineering, Nueva Granada Military University, Bogota 49300, Colombia; oscar.reyes@unimilitar.edu.co; 4LASFALTO, Guadalajara, México, 3042 Agua Marina Street, Fracc. Agua Blanca, Zapopan 45236, Mexico; omar.adame@lasfalto.com.mx

**Keywords:** fillers, stone mastic asphalt, semicircular bending, fracture energy, energy index

## Abstract

In most cases, stone mastic asphalt (SMA) mixtures placed in thin layers and subjected to stress develop early cracks (potentially resulting from being improperly affixed to the underlying layer, placed over previously cracked asphalt pavement, or placed over Portland cement concrete slabs). However, the filler used in SMA production is very influential on the performance of the mix. Fillers used in this type of mixture have a low plastic index or are inert (calcium carbonate or lime), so it is important to understand the effect of each material on the possible fissuring and cracking process of the SMA mixture. The objective of this study is to present an evaluation of the behavior of SMA asphalt mixtures with different types of filler and at different temperatures using the semicircular bend (SCB) fracture energy test. This research compares results between fracture energy and different types of filler in SMA asphalt mixtures at temperatures ranging from −10 to 25 °C.

## 1. Introduction

The main roads are constructed of flexible pavements with asphalt surfaces. However, alternative mixtures, such as stone mastic asphalt (SMA), have been adopted to improve shear strength and durability [1]. Stone mastic asphalts have proven to be reliable because they produce a significant (20%–30%) increase in stiffness and durability of the pavement compared to conventional mixtures [2]. These mixtures contain discontinuous grain-size distributions and use cellulose, natural and mineral fibers as a stabilizing agent [3], which modifies the mechanical properties of the mixtures [4]. The mixture components include coarse aggregates, a high binder content and a high filler content [2]. The most commonly used fillers are crushed stone, cement and lime [4]; however, recycled materials [5], industrial products, waste materials [6] and even polyethylene terephthalate (PET) have also been used [2].

Based on previous work, Shafiei and Namin [4], evaluated the effect of different percentages of hydrated lime filler (in the dry state and not as an additive) on the performance and mechanical properties of SMAs. They concluded that a percentage greater than 5% changes the trend and reduces rutting indexes. On the other hand, Pereira, Freire, Sá da Costa, Antunes, Quaresma and Micaelo [6] analyzed the ductility of SMAs through strength-ductility tests. For this, they manufactured samples with the same type of paving-grade bitumen but with different types and concentrations of filler. They showed that the ductility is related to the type and concentration of the filler. Finally, Topini, Toraldo, Andena and Mariani [5] revised the compaction properties, volumetric characteristics, and mechanical behavior of SMAs; the mechanical behavior was evaluated with the indirect tensile test at different temperatures. For this, they used two types of recycled material used as filler (stabilized bottom ashes and electric arc furnace steel slag), and their results showed that the behavior of these fillers was similar to or better than that of mixtures with a conventional filler.

Therefore, the objective of this research is to analyze the effect of different fillers (lime, calcium carbonate (CaCO_3_), Filler 1, Filler 2, and Filler 3) on the energy index and fracture energy of SMA mixtures through semicircular bend (SCB) fracture tests at different test temperatures. These tests produced displacement-load graphs of SMAs. Furthermore, the relationship between the test temperature and the fracture energy and energy index was found. Additionally, the physicochemical properties of each filler were evaluated based on the methylene blue value, plasticity index, granulometric composition, scanning electron microscopy analysis, and chemical composition.

## 2. Background

Stone mastic asphalt mixtures were developed in Germany at the end of the 1960s by STRABAG and J. Rettenmaier. Stone mastic asphalts were intended to reduce deterioration, increase useful life and reduce maintenance costs compared to conventional pavements. However, despite their advantages, their use was not normalized in Germany until 1984, when SMAs began to be adopted in other countries in Europe, America and Asia [7,8].

Stone mastic asphalt mixtures are asphalt mixtures characterized by a large amount of coarse aggregate, a high proportion of binder and mineral powder, a low amount of intermediate-size aggregate and a small amount of stabilizing additive. These proportions generate a good mineral structure and a high proportion of filler-based mastic, which enable a high carrying capacity without affecting the flexibility of the mixture [9,10].

The SMA mixtures were conceived with clear and well-defined objectives: To increase the durability, safety and stability of communication routes and to generate savings in their construction. Stone mastic asphalts are hot-prepared mixtures characterized by being impermeable and resistant to the formation of ruts. Arabani and Ferdowsi [11], Erdlen and Yu [7], Guide [9], Kandhal [10], Hainin, Reshi and Niroumand [8], and others have described the advantages of using SMA-type mixtures (Figure 1).

Understanding the propagation of cracks and the time over which deterioration occurs in flexible pavement is the greatest challenge for researchers. Therefore, several modeling schemes exist to discover when and how failure will occur, and understanding these aspects is one of the purposes of this study.

It has been stated that damage due to cracking in an asphalt layer causes the greatest damage, because this is a reflection that some part of the pavement structure has failed [12]. In addition, the passage of vehicles across deteriorated asphalt pavement is uncomfortable for the driver, and in some cases, the repair of damaged pavement is quite expensive. As a result, different laboratory tests have been proposed to mitigate this problem and have suggested a more representative model of asphalt layer deterioration in the field [13,14,15].

Currently, methods exist to prevent this type of failure, such as the stability and Marshall flow and dynamic and static tests, which attempt to predict the deterioration and durability of an asphalt layer [12,16,17,18]. Moreover, there are relatively new tests that study fracture energy, which can provide information on the behavior and durability of an asphalt layer, such as the SCB test [11,19,20,21,22], which is the test used to evaluate the SMA mixtures in this investigation.

Although there are several methods for the study of asphalt pavement or bearing surface cracking, not all of them satisfy the necessary requirements or can accurately measure the different problems they face. Thus, researchers have crafted methods to develop satisfactory theories and have standardized the problems in laboratory settings [12,23].

The SCB test determines the fracture energy necessary to cause an asphaltic mixture to crack by measuring various physical and/or chemical characteristics and external factors, such as temperature or induced specimen damage.

The SCB test was presented by Kuruppu et al. [24] as a quick 3-point bending test. This test is used to evaluate resistance to fracturing at different temperatures and with different material characteristics, such as different aggregates, asphalt types, asphalt contents and filler types (as in the case of this investigation). In general, the standard procedure is to perform the test at three different temperatures and to compare the results for different compositions. This test uses the finite element method to determine the variation in the load intensity factor with respect to the crack length.

Despite the use of the finite element method to calculate the load with respect to the crack, the fracture energy is still calculated through the experimental results of different tests on the specimens, simply because there are factors that make the test difficult to carry out accurately [25]. For example, a rock particle oriented in the direction of the crack will result in resistance at that point. Thus, it is recommended to test at least six different samples at three different temperatures and loading speeds. The results are directly affected by the type of mixture, which includes the type of stony material, asphalt, and filler. Moreover, another factor that affects the results is the void ratio in each specimen.

Importantly, the SCB test is based on the assumption that the fracture energy is absorbed only by the affected area where the fracture occurs and does not affect the rest of the sample, although recent studies have shown that the entire sample is affected [26,27]. However, this assumption is valid in this test, because the purpose of conducting a test using the SCB method is to measure fracture energy, which means that the goal is only to determine the energy required at different temperatures and loading speeds to produce the fracture.

The SCB test has been used for various purposes, including evaluating the behavior of dense asphalt mixtures with two types of gradations at temperatures of 0 and 25 °C by Arabani and Ferdowsi [11], assessing different types of asphalt and ages by Kim et al. [28] and measuring the stress intensity factor (K_IC_) in different types of asphalt and at different test temperatures by Pszczola and Szydlowski [29]. Furthermore, the results derived from the SCB test have been compared with those obtained through the indirect tensile test (IDT) by Kim, Mohammad and Elseifi [28] and the uniaxial tension stress test (UTST) and bending beam test (BBT) by Pszczola and Szydlowski [29], who showed that the SCB test results and the UTST and BBT results showed excellent correlations.

## 3. Materials and Methods

To fulfill the objective of this research work, the behavior of SMA mixtures manufactured with asphalt PG 70-16 and five different fillers were analyzed. The SMA mixtures were manufactured by combining the asphalt with different fillers (lime, CaCO_3_, Filler 1, Filler 2, and Filler 3) with a filler/asphalt proportion of 1.51. Additionally, 4 specimens of each SMA mixture were manufactured and tested to obtain an average.

To perform the analysis, it was necessary to determine the characteristics of the aggregates (coarse and fine), the fillers, the asphalt binder, and the SMA mixtures. The specimens were compacted using a Superpave gyratory compactor (Matech, Treviolo, Italy) at 100 gyres with an internal angle of 1.16° and a stress of 600 kPa. Subsequently, the behavior analysis of the different SMA mixtures was conducted via the SCB fracture test at temperatures of −10, 5, 15 and 25 °C. During each test, the load was controlled by displacement at a constant speed of 1 mm/min.

### 3.1. Aggregate Characteristics

The aggregate used was of basaltic origin and was 100% produced via crushing. Additionally, two gradations of material were used: Gravel with a maximum size of 19 mm and sand with a maximum size of 4.75 mm. The sampling of the aggregate material was conducted in accordance with the SCT regulation M-MMP-4-04/02, the characteristics of which are shown in Table 1 and Table 2.

Additionally, the grain size distribution of coarse and fine aggregates is shown in Figure 2. It is evident that the particle size distribution is within the limits established by current regulations. The design was carried out in accordance with the AASHTO MP-8 standard, in which a grain size distribution adjusted to the limits for a nominal maximal size of 9.5 mm was established.

### 3.2. Physicochemical Properties of Fillers

In the manufacture of the test SMA mixtures, a 1.51 asphalt/filler proportion was used. Five types of fillers were used for the formation of the test SMA mixtures.

The physicochemical properties of the fillers were analyzed at the Laboratorio de Microscopia de la Universidad Nacional Autónoma de México (see Table 5, Figures 5 and 6). The properties analyzed are as follows:Absorption of methylene blue (TC-Technologies, Puebla, Mexico): This test measures the amount of undesirable clay present in the filler; high values (>13 mg/g) are associated with laminar particles with a large specific surface that react in the presence of water, which activates its expansive potential (deleterious).Plasticity index (ALCON, Guadalajara, Mexico): This test assesses the plasticity properties of the clay contained in the filler, and plastic index values greater than 4 are considered unsuitable and associated with laminar-shaped particles.Granulometric composition (ALCON, Guadalajara, Mexico): The particle size distribution is obtained, including the percentage of colloidal materials smaller than 0.002 mm, which are considered detrimental due to their expansion potential.Analysis by scanning electron microscopy (JEOL USA Inc., Peabody, MA, USA): The micrographs reveal the shapes (equidimensional or laminar) and sizes of the particles; therefore, the results of methylene blue and plasticity index can be verified.Chemical analysis (Surfax, Zapopan, Mexico): The aim of this test is to determine the chemical elements contained in each filler.

### 3.3. Asphalt Binder Characteristics Used in Test SMA Mixtures

The optimal asphalt content (AC) was calculated using a Superpave gyratory compactor at 100 gyres, with an internal angle of 1.16° and a stress of 600 kPa. The results of the analysis of the asphalt used in the test SMA mixtures are shown in Table 3.

Additionally, the volumetric properties of the SMA mixtures, such as gravity mixture maximum (G_mm_), gravity mixture bulk (G_mb_), air voids, voids mineral aggregate (VMA), and voids filled asphalt (VFA), were obtained. These properties are shown in Table 4.

### 3.4. Semicircular Bend Fracture Test

The SCB fracture test has been used to obtain the fracture toughness, fracture energy, and stress-softening curves of asphaltic materials. The SCB fracture test is simple to perform and allows one to prepare test specimens easily through Superpave gyratory compactor or field coring. In addition, a mixed-mode fracture can be achieved by modifying the geometry by changing the length (a) and angle of notch (α), as well as the length of the support gap (2s) (see Figure 3a), as mentioned by Ban et al. [30].

The specimens were made with the same proportions of Asphalt Content (AC), grain size distribution, and type of stone aggregate but had different types of filler. The optimum dosage of filler was determined from calculation to G_mm_ and G_mb_, as illustrated in Table 4. The specimens were 4 cm in diameter with a 1 cm groove depth, and the SCB tests were controlled through Critical tip opening displacement (CTOD) and were performed at a constant loading speed of 1 mm/min and at four different temperatures (−10, 5, 15 and 25 °C).

Generally, the SCB test yields the stress intensity factor (K_IC_), critical strain energy (J_c_) or fracture energy (*G_D_*). However, in the present study, the mechanical properties of the SMAs (see Table 6) were estimated from the expressions (1) to (7) derived from the SCB test and the displacement-load relationship (see Figure 4):(1)$$R_{T}=\frac{1000\times F_{max}}{h\times l}$$
where $R_{T}$ is the tensile strength, $F_{max}$ is the work performed up to the maximum load, *h* is the specimen thickness and *l* is the original ligament length.

The tensile stiffness index $I_{RT}$ is calculated by:(2)$$I_{RT}=\frac{\frac{1}{2}\times F_{max}}{\Delta_{m}}$$
where $\Delta_{m}$ is the displacement before the maximum load to ½ $F_{max}$. The calculation of the dissipated energy during cracking ($G_{D})$ is determined by:(3)$$G_{D}=\frac{W_{D}}{h\times l}$$
where the work done in the cracking process is calculated ($W_{D}$) by:(4)$$W_{D}=\overset{n}{\underset{i=1}{\sum }}\left(x_{i+1}-x_{i}\right)y_{i}+0.5\left(x_{i+1}-x_{i}\right)\left(y_{i+1}-y_{i}\right)$$
where $x_{i}$ is the displacement recorded, $y_{i}$ is the load recorded and n is the point at which the load decreases to 0.1 kN.

The tenacity index ($I_{T}$) is obtained by the following equation:(5)$$I_{T}=\frac{W_{D}-W_{Fmax}}{h\times l}(\Delta_{mdp}-\Delta_{Fmax})$$
where $W_{Fmax}$ is the work performed up to the maximum load (prepeak area), $\Delta_{mdp}$ is the displacement at 50% postpeak load, and $\Delta_{Fmax}$ is the displacement at the maximum load (see Figure 4).

The calculation of the energy index (*U*) is determined by:(6)$$U=\frac{W_{S}}{h\times l}\left(\frac{W_{S}}{W_{D}}\right)$$
where $W_{S}$ is the work done in the postpeak area (softening zone) is determined by:(7)$$W_{S}=W_{D}-W_{Fmax}$$

## 4. Results and Discussion

Derived from the physicochemical analysis for different fillers, Table 5 shows that, based on the methylene blue absorption test, Filler 2 and Filler 3 are not suitable for improving SMA mixtures, while lime, CaCO_3_ and Filler 1 presented methylene blue values of less than 5 mg/g. Therefore, these SMA mixtures will have a better performance, according to the recommendation AMAAC RA-05 [31].

Additionally, the plasticity index shown in Table 5 also shows that lime, CaCO_3_ and Filler 1 do not exhibit plasticity; therefore, they are recommendable for use in SMA mixtures. In contrast, Filler 2 and Filler 3 present medium to high plasticity indexes; therefore, these fillers should not be used.

However, analysis of the granulometric composition of the different fillers revealed that the industrial-product fillers (lime and CaCO_3_) are composed mostly of finer particles and have discontinuous grain size distributions, particularly the lime filler. Meanwhile, the material with the most continuous grain size distribution is Filler 1 (see Figure 5).

Figure 6 shows that the particles of the industrial-product fillers (lime and CaCO_3_) have rounded shapes (Figure 6a,b). Additionally, Figure 6c,d shows that the particles of Filler 1 and Filler 2 are cubic in shape, whereas Filler 3 tends to exhibit needle-like or laminar forms (Figure 6e).

The chemical compositions of the industrial-origin fillers are characterized primarily by oxygen and calcium. In contrast, the crushed rock fillers have more chemical elements; specifically, Filler 3 presents a high combination of aluminum and silicon, which is characteristic of clay materials (Figure 6).

Figure 7a (according to the SCB test at −10 °C) allows determination of which SMA mixture requires greater fracture energy, because deterioration from thermal cracking can occur under these temperature conditions.

All results of the SCB test at −10 °C are shown in Table 6 and Figure 7. It is evident that the SMA mixture with Filler 1 has the highest values, with 1215 and 1540 J/m^2^ for the test temperature-deformation energy (*U*) and test temperature-fracture energy (*G_D_*), respectively. These results imply that this mixture presents the best behavior. Moreover, the lowest values in the test were associated with the SMA mixture with Filler 2; consequently, this mixture is more likely to be fragile (see Table 6).

Furthermore, in the SCB test at 5 °C (see Figure 7b), the best mixture behavior was presented by Filler 1, registering the highest values of *U*, *G_D_* and *I_T_*, corresponding to 1099, 2379 and 979 J/m^2^, respectively. The lowest results were associated with the SMA mixture with lime filler. Therefore, Filler 1 has the greatest flexibility and tenacity at 5 °C, which means that thermal fissures and transition to fatigue cracking are less likely to occur (see Table 6).

Figure 7c (SCB test at 15 °C) clearly shows that the SMA mixture tends to exhibit a viscoelastic response due to the rheological properties of the asphalt. In this test, the best behavior is recorded with the mixture containing Filler 1, with values of 241, 494 and 182 J/m^2^ for *U*, *G_D_* and *I_T_*, respectively (see Table 6). Therefore, Filler 1 has greater resistance to fatigue cracking that occurs under these temperature conditions. The minimal recorded values correspond to the CaCO_3_ and lime fillers.

The SCB test results at 25 °C are shown in Figure 7 and Table 6, and the results of the SCB test are presented. The highest records are recorded in the mixtures with Filler 1 (values of 309, 762 and 484 J/m^2^ for *U*, *G_D_* and *I_T_*, respectively) and CaCO_3_ (values of 367, 832 and 305 J/m^2^ for *U*, *G_D_* and *I_T_*, respectively). The lowest results are for Filler 2, Filler 3 and lime. Therefore, Filler 1 and CaCO_3_ have the highest flexibility and tenacity at 25 °C, which makes fatigue cracking less likely to occur in these fillers.

According to Table 6, in the SCB test at −10 °C, it is evident that the SMAs resist forces of greater magnitude that produce little displacement, with which the rigidity indexes (*I_RT_*) increase and become more fragile; therefore, they have tenacity index (*I_T_*) values of zero. In the tests at 5 °C, 15 °C and 25 °C, the SMAs show increasing tenacity index values, indicating that the mixtures become more resistant to loads and deformation. Based on the above results, Filler 1 presents the highest tenacity index values and is therefore the most tenacious and least fragile SMA mixture; therefore, more energy is required to deform and fracture this SMA mixture.

In Figure 8 and Figure 9, Filler 1 is shown to have greater deformation and fracture energy values at all testing temperatures, which shows that the SMA mixtures with this filler are more resistant to the cracking process. Additionally, the lime filler and Filler 3 tend to show low index energy and fracture energy values at the different temperatures in the SCB test. On the other hand, the CaCO_3_ filler and Filler 2 present greater variability in the index energy and fracture energy values, which confirms their instability in the thermal cracking and fatigue cracking processes.

Consequently, it was determined that Filler 1, derived from crushed rock, has the best behavior in terms of the energy fracture parameters measured through the SCB test, guaranteeing better resistance to possible thermal or fatigue cracking than the other fillers. In addition, the results demonstrated that industrial products (particularly lime) are more susceptible to cracking.

## 5. Conclusions

The aim of this work was to evaluate the effect of fillers on the index energy and fracture energy of SMAs with different fillers. Thus, five different fillers were mixed with one asphalt type at the same filler-to-asphalt ratio and tested using the SCB fracture test. The deformation and fracture energy results were analyzed and related to the displacement-load relationship and physicochemical composition results in terms of filler-asphalt interaction.

The effect of fillers in the displacement–load curves showed that the resistance of the SMA mixtures depends on the test temperature. However, with a temperature of 25°C the fracture energy and index energy increase. Analyzing the physical and chemical properties of the fillers, it is evident that the dosage, the shape of the particles and the chemical composition affect the performance of the SMA mixture.

Thus, the best performance of the SMA was obtained with fillers with the lowest proportion of aluminum and silicon (harmful clays components). Likewise, if the filler particles are equidimensional (cubic, spherical or similar), they have a smaller specific surface area, absorb less methylene blue, and have lower plasticity index values. Thus, the reduction of these properties improves the performance of the SMA. For that reason, Filler 1 and lime filler performed better than other fillers at lower temperatures, whereas Filler 1 showed lower performance at intermediate temperatures. 

Additionally, the relationship between the temperature-energy index and fracture energy indicates that lower proportions of aluminum and silicon are associated with better performance; therefore, more energy is required to produce deformation and a fracture when Filler 1 is added to the SMA mixture. In contrast, the lime filler presents the worst performance with respect to the energy index and fracture energy.

According to the mechanical properties, Filler 1 presents the highest tenacity index values in all the tests at different temperatures. Therefore, the SMAs mixed with Filler 1 are more tenacious, more ductile and less fragile than the other mixtures; therefore, more energy is required to deform and fracture this SMA mixture. In contrast, the lime filler has the lowest tenacity index values and is therefore the one that presents the most unfavorable behavior.

Importantly, the use of industrial materials such as CaCO_3_ and lime does not always guarantee good performance of the SMA mixtures in the cracking process, as evidenced by the fillers evaluated with the SCB test in this study.

## Figures and Tables

**Figure 1 materials-12-00288-f001:**
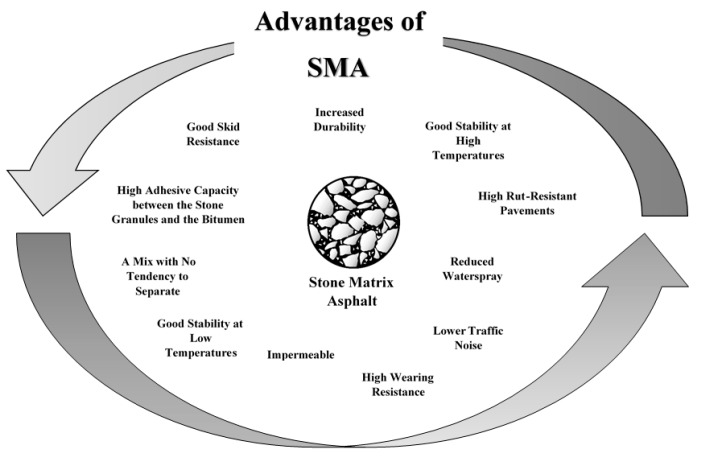
Stone mastic asphalt (SMA) advantages.

**Figure 2 materials-12-00288-f002:**
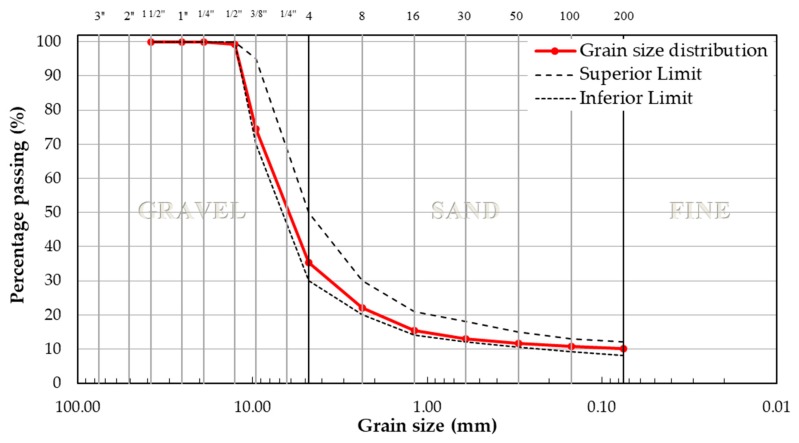
Grain size distribution of aggregates using the AASHTO MP-8 procedure.

**Figure 3 materials-12-00288-f003:**
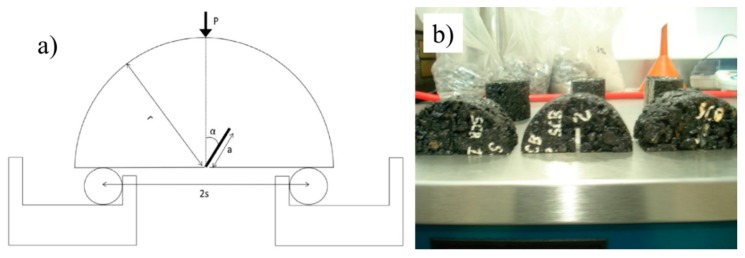
(**a**) Schematic configuration of the semicircular bend (SCB) test and (**b**) geometry of the SCB specimens.

**Figure 4 materials-12-00288-f004:**
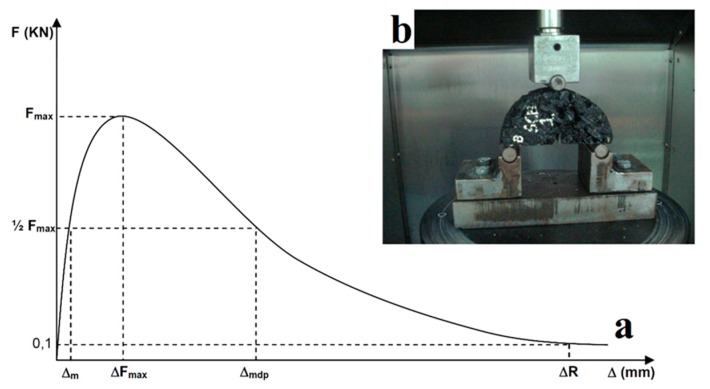
(**a**) Load–displacement curve and (**b**) sample used in the test SCB. *W_D_* = the work done in the cracking process (F-Δ area under the load–displacement curve); *W_FMax_* = the work done up to the maximum load (prepeak area); *W_S_* = the work done in the softening zone (postpeak area).

**Figure 5 materials-12-00288-f005:**
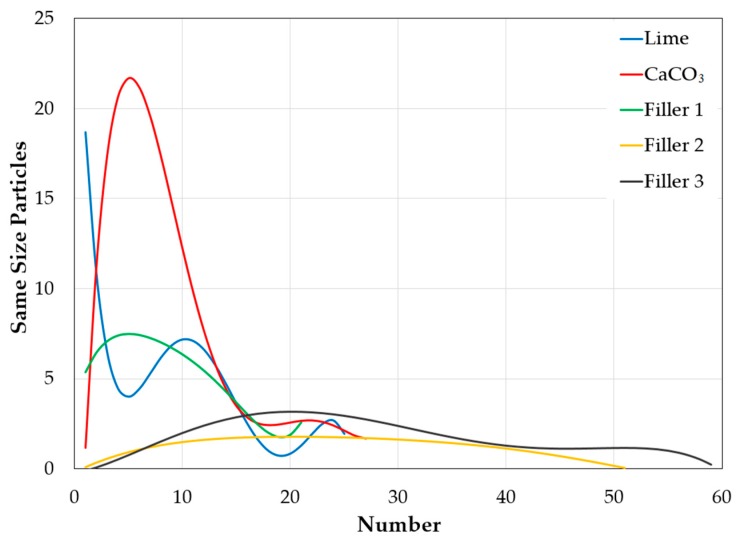
Granulometric composition of lime, CaCO_3_, Filler 1, Filler 2, and Filler 3.

**Figure 6 materials-12-00288-f006:**
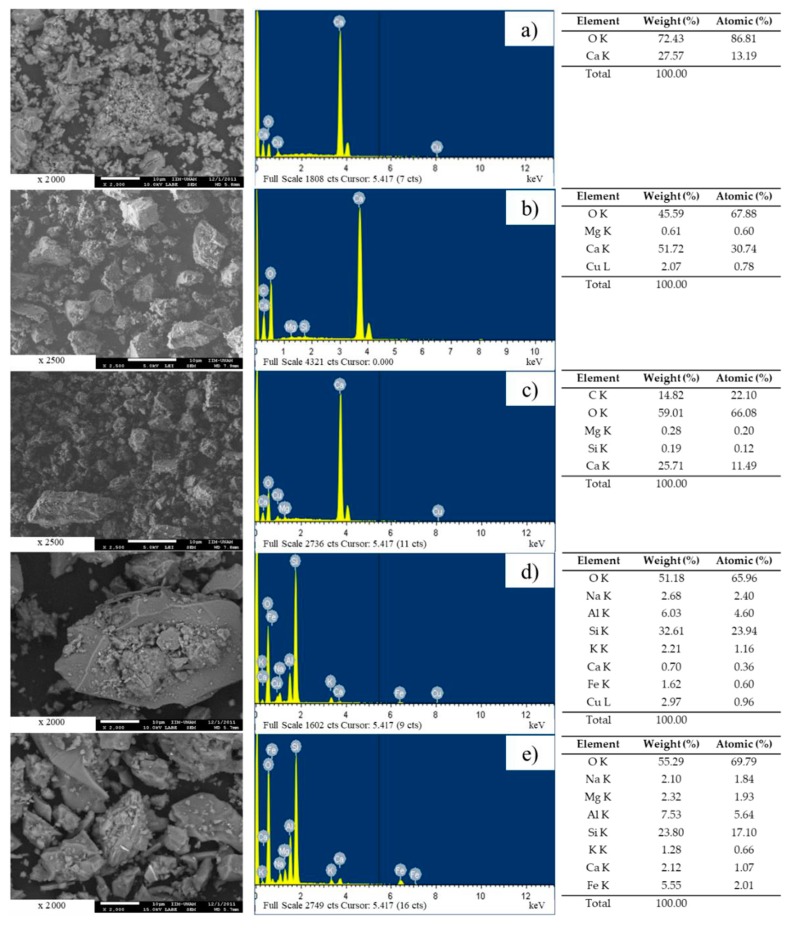
Scanning electron microscopy images of (**a**) lime; (**b**) CaCO_3_; (**c**) Filler 1; (**d**) Filler 2 and (**e**) Filler 3.

**Figure 7 materials-12-00288-f007:**
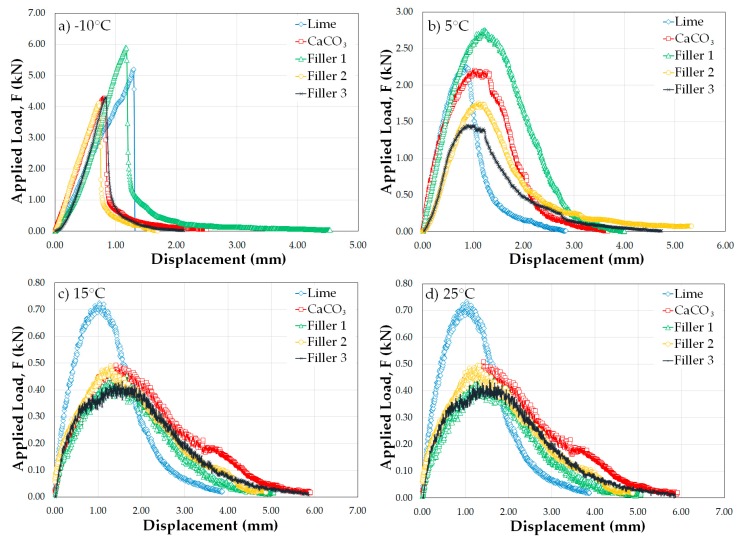
Semicircular bend load–displacement graph of all fillers at (**a**) −10 °C; (**b**) 5 °C; (**c**) 15 °C and (**d**) 25 °C.

**Figure 8 materials-12-00288-f008:**
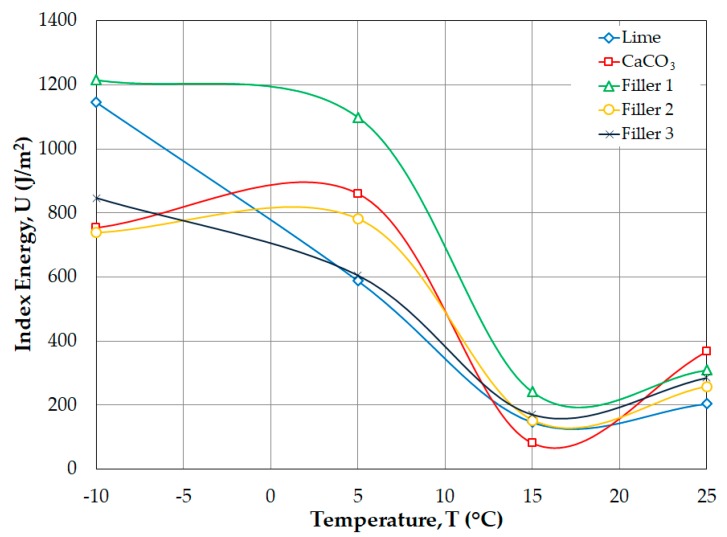
Index energy (*U*) as a function of test temperature.

**Figure 9 materials-12-00288-f009:**
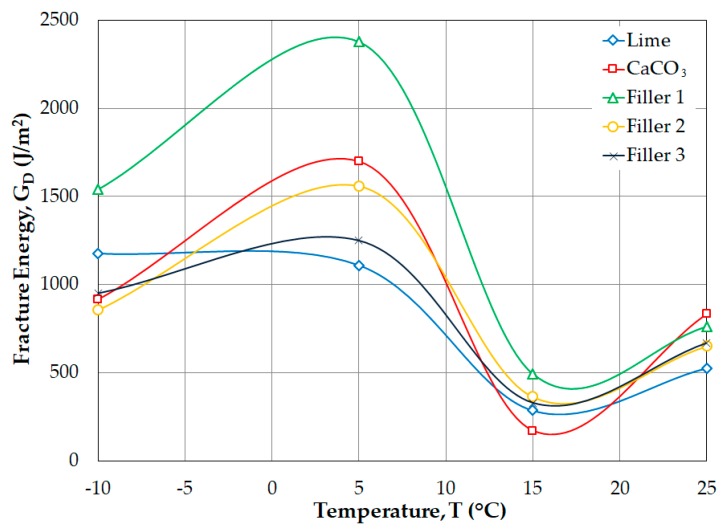
Fracture energy (*G_D_*) as a function of test temperature.

**Table 1 materials-12-00288-t001:** Characteristics of the coarse aggregate.

Characteristic	Normative	Value Obtained	Specification PA-MA-001/2008
Abrasion and Impact in the Los Angeles Machine, %	ASTM C131-03	13%	30 max. (structural layers)
Percentage of Fractured Particles in Coarse Aggregate, % (2 faces or more)	ASTM D 5821	100%	90 min.
Elongated Particles in Coarse Aggregate, %	ASTM D 4791	24%	3 a 1%, 15 max.
Flat Particles in Coarse Aggregate, %	ASTM D 4791	16%	3 a 1%, 15 max.
Specific Gravity of Coarse Aggregate	ASTM C127-07	2.72	-
Absorption of Coarse Aggregate	ASTM C127-07	0.75%	-

**Table 2 materials-12-00288-t002:** Characteristics of the fine aggregate.

Test	Normative	Value Obtained	Specification PA-MA-001/2008
Equivalent Value of Fine Aggregate, %	ASTM D 2419	61	50 min. (structural layers)
Methylene blue, mg/g	Recommendation AMAAC RA-05/2010	10	15 max. (structural layers)
Specific Gravity of Fine Aggregate	ASTM C128-04	2.55	-

**Table 3 materials-12-00288-t003:** Asphalt analysis.

Binder Analysis	Test	Result
Original binder	Penetration at 25 °C 100 g 5 s (1/10 mm)	69
	Elastic recovery by torsion at 25 °C (%)	5
	Softening point at 5 °C/min. (°C)	49
	Performance grade, PG	70
	Cleveland flash point	>260
	Brookfield viscosity at 135 °C sc4-27 12 rpm (cP)	530
	Module DSR to PG (G*/senδ) (kPa)	1.21
Aged binder RTFO	Loss mass at 163 °C (%)	0.57
	Performance grade, PG	70
	Module DSR to PG (G*/senδ) (kPa)	2.23
Aged binder PAV	Module DSR at 34 °C (G*senδ) (kPa)	2102
	Slope (m) BBR test at −6 °C	0.312
	Module stiffness BBR test at −6 °C (MPa)	287

DSR: Dynamic Shear Rheometer; RTFO: Rolling Thin-Film Oven; PAV: Pressure Aging Vessel.

**Table 4 materials-12-00288-t004:** Volumetric properties of the SMA mixtures.

Filler Type	AC (%)	G_mm_	G_mb_	Air Voids (%)	VMA (%)	VFA (%)
Lime	6.5	2.394	2.298	4.0	18.2	78.0
CaCO_3_	6.5	2.394	2.298	4.0	18.2	78.0
Filler 1	6.5	2.393	2.299	3.9	18.3	78.6
Filler 2	6.5	2.394	2.300	3.9	18.3	78.6
Filler 3	6.5	2.393	2.299	3.9	18.3	78.6

**Table 5 materials-12-00288-t005:** Methylene blue absorption, plasticity index, and pH values of analyzed fillers.

Filler Type	Methylene Blue Value (mg/g)	Performance Recommendation AMAAC RA-05	Plasticity Index (%)	Early Performance	Potential of Hydrogen, pH
Lime	1	Excellent	No plasticity	No plasticity	12.80
CaCO_3_	3	Excellent	No plasticity	No plasticity	11.60
Filler 1	4	Excellent	No plasticity	No plasticity	10.60
Filler 2	17	Problems/possible failure	5.20	Medium plasticity	9.30
Filler 3	32	Failed	9.40	High plasticity	7.50

**Table 6 materials-12-00288-t006:** The results for all fillers analyzed in the SCB test at −10 °C, 5 °C, 15 °C and 25 °C.

**Filler Type**	**−10 °C**							**5 °C**						
	**Δ*F_max_***	**Δ*F_max_***	**Δ*R***	***I_RT_***	***U***	***G_D_***	***I_T_***	**Δ*F_max_***	**Δ*F_max_***	**Δ*R***	***I_RT_***	***U***	***G_D_***	***I_T_***
	**(kN)**	**(mm)**	**(mm)**	**(kN/mm)**	**(J/m^2^)**	**(J/m^2^)**	**(J/m^2^)**	**(kN)**	**(mm)**	**(mm)**	**(kN/mm)**	**(J/m^2^)**	**(J/m^2^)**	**(J/m^2^)**
Lime	4.72	0.95	0.95	5.59	1146	1176	n.a.	2.30	0.83	2.17	3.93	588	1109	190
CaCO_3_	4.27	0.69	1.35	7.13	753	914	n.a.	2.28	1.14	2.89	3.11	860	1699	487
Filler 1	4.91	0.99	2.46	5.70	1215	1540	n.a.	2.85	1.28	3.46	2.95	1099	2379	979
Filler 2	4.44	0.74	0.94	6.13	738	855	n.a.	2.39	1.06	2.66	3.20	782	1559	344
Filler 3	4.63	0.85	0.78	5.97	847	949	n.a.	1.83	1.04	2.61	3.06	605	1249	403
**Filler Type**	**15 °C**							**25 °C**						
	**Δ*F_max_***	**Δ*F_max_***	**Δ*R***	***I_RT_***	***U***	***G_D_***	***I_T_***	**Δ*F_max_***	**Δ*F_max_***	**Δ*R***	***I_RT_***	***U***	***G_D_***	***I_T_***
	**(kN)**	**(mm)**	**(mm)**	**(kN/mm)**	**(J/m^2^)**	**(J/m^2^)**	**(J/m^2^)**	**(kN)**	**(mm)**	**(mm)**	**(kN/mm)**	**(J/m^2^)**	**(J/m^2^)**	**(J/m^2^)**
Lime	0.74	0.58	1.18	2.33	146	286	56	0.76	0.87	2.40	1.43	204	526	159
CaCO_3_	0.31	0.78	1.49	1.55	81	170	58	0.62	1.80	4.37	0.48	367	832	305
Filler 1	0.66	1.05	2.37	1.65	241	494	182	0.60	1.63	3.91	0.64	309	762	484
Filler 2	0.62	0.71	1.76	2.16	153	363	130	0.60	1.25	3.17	0.74	258	650	287
Filler 3	0.58	0.85	1.69	1.63	170	328	87	0.60	1.67	3.81	0.60	285	668	323

Note: *F_max_* = maximum load, Δ*F_max_* = maximum load displacement, Δ*R* = break displacement, *I_RT_* = stiffness index to *F_max_*, *I_T_* = tenacity index, *G_D_* = fracture energy, and *U* = index energy.

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
