# Peer review of "Analysis of the Behavior of SMA Mixtures with Different Fillers Through the Semicircular Bend (SCB) Fracture Test"

_materials, 2019, doi:10.3390/ma12020288_

Round 1
Reviewer 1 Report
Comments and Suggestions for Authors:
In this paper the influence of different types of fillers used in SMA asphalt mixture on results from SCB test method was evaluated and discussed. The reviewer would like to thank the authors for their efforts. Authors are suggested to address some comments, as follows:
1. The paragraph Introduction must be improved. The dominant part of that paragraph and also Fig 1 focused on SMA mixture technology and advantages of SMA. But the true is that SMA technology is not a new idea. The Authors should consider to add more discussion on SMA properties according to SMA composition (fillers, bitumen types, aggregates) in case of resistance to fracture. The SCB test method is a tool to assess the fracture toughness.
2. Please consider to discuss about the bitumen/filler ratio as a parameter that influence on stiffness of asphalt mixture and as a consequence on fracture properties.
3. Please add the Objectives of the paper (at the end of Introduction paragraph or separate paragraph before Experimental analysis.
4. It is recommended to have the manuscript review of English language. Please decide the use and meaning of descriptions: bitumen/asphalt/asphalt mixture. If the Authors use all of the descriptions it is important to write clear explanation.
5. Table 2: according to Original binder properties there are twice information about Performance grade PG.
6. Figure 2: Please add the SI description of sieves. Now it is only in inches – US description. Please add millimetres.
7. Figure 3: Please improve the quality of figures (right side of Fig. 3. It is hard to read the description of materials.
8. SCB test method is based simply on determination of asphalt mixture resistance to fracture KIC, which is calculated on the basis of the maximum force recorded during bending of the specimen. The fracture toughness KIC was estimated, using the following equation:
where: a is the notch depth, σ0 test extreme stress, YI normalized stress intensity factor due to type I fracture. That description is according to the original test method described by in the EN 12697-44 standard and was applied in many literature sources. Please try to explain any differences between SCB test methods.
9. Please consider to reorganize the Conclusions. It should be clearly connected to the research findings.

Reviewer 2 Report
The paper is interesting however, it is not clear why the crack opening was not controlled with a CMOD. No post peak curve is observed for low temperature (-10C), missing important information on fracture energy. This would be recommended. Please, explain in the paper.
More details are needed in terms of literature review, experimental method and conclusions. Please, extend the paper.
Reviewer 3 Report
The paper describes SCB testing carried out on SMA mixtures prepared with different mineral fillers.
The paper structure does not conform to usual standards: Introduction, Materials and Methods, Results and Discussion, Conclusion. Section numbering jumps from 2 to 4.
Given the topic of the paper, one would expect a review of literature on the effect of filler on SCB test results, or on the effect of filler on SMA properties. Instead the introduction is very generic (figure 1 is suitable for a book not for a scientific paper) and contains several English language errors and misuse of technical terms.
Although the paper focuses on filler, the main physical properties of fillers were not measured or reported (e.g. grading distribution, Rigden voids, pH etc). Chemical compositions should be reported in terms of oxides.
The experimental campaign i limited. The authors should use SI units and give a better description of the specimens and how they were prepared (h = ?). There are North American (ASTM, AASHTO) and European (EN) standards on SCB testing of bituminous mixtures, the Authors should consult those standards. What is the reason for using the parameters described in equations 1, 2, 5, 6?
The plots in figure 5 are too small and not readable.
The authors repeatedly refer to "fatigue" and "elastic behaviour" without a real reason. "Fatigue resistance" is different from "cracking resistance" and there is no elastic response in the test results (except probably at -10°C). I think probably the author should refer to "viscoelastic" response.
The second part of the Conclusions section (from line 205 to 213) is not a conclusion, but again contain generic statements which do not follow from the presented results.
Round 2
Reviewer 1 Report
The reviewer would like to thank the authors for all the corrections and improvements of the paper. It improves the quality and scientific level of the paper. Authors are suggested to address some comments, as follows:
1. The paragraph Introduction must be improved. The dominant part of that paragraph and also Fig 1 focused on SMA mixture technology and advantages of SMA. But the true is that SMA technology is not a new idea. The Authors should consider to add more discussion on SMA properties according to SMA composition (fillers, bitumen types, aggregates) in case of resistance to fracture. The SCB test method is a tool to assess the fracture toughness – DONE, thank you.
2. Please consider to discuss about the bitumen/filler ratio as a parameter that influence on stiffness of asphalt mixture and as a consequence on fracture properties – DONE, thank you.
3. Please add the Objectives of the paper (at the end of Introduction paragraph or separate paragraph before Experimental analysis – DONE, thank you.
4. It is recommended to have the manuscript review of English language. Please decide the use and meaning of descriptions: bitumen/asphalt/asphalt mixture. If the Authors use all of the descriptions it is important to write clear explanation – DONE, thank you.
5.Table 2:according to Original binder properties there are twice information about Performance grade PG – please check.
6. Figure 2: Please add the SI description of sieves. Now it is only in inches – US description. Please add millimetres – DONE, thank you.
7.Figure 3: Please improve the quality of figures (right side of Fig. 3. It is hard to read the description of materials – DONE, thank you.
8. SCB test method is based simply on determination of asphalt mixture resistance to fracture KIC, which is calculated on the basis of the maximum force recorded during bending of the specimen. The fracture toughness KIC was estimated, using the following equation:
where: a is the notch depth, σ0 test extreme stress, YI normalized stress intensity factor due to type I fracture. That description is according to the original test method described by in the EN 12697-44 standard and was applied in many literature sources. Please try to explain any differences between SCB test methods – DONE, thank you.
9.Please consider to reorganize the Conclusions. It should be clearly connected to the research findings – DONE, thank you.
Reviewer 2 Report
The authors only partially replied to the question on the use of CMOD. As this is an experimental issue, they should clearly state in the paper why they didn’t use a CMOD. It is obvious that the material is brittle at lower temperatures.
Reviewer 3 Report
The paper was improved but still contain several issues and need a further review. Consider for example the very first row: “Many RUNWAYS, taxiways and even airport RUNWAYS …”.
P1L37. Not only cellulose (i.e. natural) but also mineral fibres. Fibres are used mainly to prevent drain-down phenomena. The National Center for Asphalt Technology (NCAT) published several interesting reports on SMA.
L142 “graduations” should be “gradations”
The gradations limits shown in Figure 2 are strange. This mixture does not look like a SMA mixture which has to be gap-graded. Instead this is clearly continuously-graded. I wonder what kind of regulation was followed.
In Table 2 values of Gmm and Gmb are not clear. Usually specific gravity is dimensionless.
L136, L174, L185 please use uniform terminology.
Is Ws = WD – Wfmax?
Results shown in Table 5 demonstrate that Filler 2 and Filler 3 should not be used. Why did you use these fillers?
Figure 5: a single plot with 5 curves would be much more readable.
Please use only “.” As decimal separator.
L191 “The specimens were made with the same proportions of AC, grain size distribution, and type of stone aggregate but had different types of filler, as illustrated in Figure 5.” A comment on this methodological approach. Since you change filler type but keep the same dosage it is certain that the dosage is not “optimum”. Therefore, the results are better for those filler for which the chosen dosage was optimal or close to optimal.
Conclusions should be more concise and reflect the actual conclusions of this study. For example, the fact that “resistance of the SMA mixtures depends on the test temperature” is well known.
L320 Why “in general”? I really believe the opposite is true: in general industrial products can perform better than natural, provided they are employed in the correct way.
Round 3
Reviewer 3 Report
The Authors corrected the manuscript according to the comments.
"The optimum asphalt for each type of filler was obtained through the volumetric behavior as shown in table 4."
From table 4, the same asphalt content was used for all the mixtures. Hence, apparently, the filler type did not affect the volumetric properties. This is quite surprising.